# Lactic Bacteria with Plant-Growth-Promoting Properties in Potato

Lilian Dutra Panetto [1], Joyce Doria [2], Carlos Henrique Barbosa Santos [1], Edvan Teciano Frezarin [1], Luziane Ramos Sales [1], Luana Alves de Andrade [1] and Everlon Cid Rigobelo [1,*]

1   Agricultural and Livestock Microbiology Graduate Program, School of Agricultural and Veterinary Sciences, São Paulo State University (UNESP), São Paulo 14887-900, Brazil
2   Department of Agriculture, Federal University of Lavras (UFLA), P.O. Box 3037, Lavras 37200-900, Brazil
*   Correspondence: everlon.cid@unesp.br

**Abstract:** This study aimed to evaluate the abilities of three bacteria, *Bacillus cereus*, *Succinovibrio dextrinosolvens*, and *Lactobacillus acidophilus*, to fix nitrogen, solubilize phosphorus, and produce cellulosic and amylolytic enzymes. Then, these bacteria were evaluated in potato plants under field conditions. The bacterium *B. cereus* showed the ability to synthesize amylase, indole acetic acid (IAA) production of 9.08 µg mL$^{-1}$, phosphorus solubilization of 14.93 mg P L$^{-1}$, and nitrogen fixation of 0.7 mg of nitrogen L$^{-1}$. *S. dextrinosolvens* showed the ability to synthesize siderophores and amylase, IAA production of 10.25 µg mL$^{-1}$, phosphorus solubilization of 41.38 mg P L$^{-1}$, and nitrogen fixation of 0.42 mg N L$^{-1}$. *L. acidophilus* showed the ability to synthesize siderophores, IAA production of 7.25 µg mL$^{-1}$, phosphorus solubilization of 5.58 mg P L$^{-1}$, and nitrogen fixation of 0.5 mg N L$^{-1}$. Some plant parameters were increased as shoot dry matter by *B. cereus*, and the mixture of bacteria increased shoot and root dry matter and increased phosphorus from the root. More studies are needed to deepen the understanding of the potential of these bacteria; however, *B. cereus* showed great potential to be used as a plant growth promoter in potato crops in the future.

**Keywords:** *Bacillus cereus*; chemical fertilizer; *Lactobacillus acidophilus*; potato





## 1. Introduction

Potato crops have tremendous importance due to the large amount of starch that is the food base for many people worldwide. For its production, potato requires a large amount of chemical fertilizer, and many studies relate the high use of chemical fertilizers with diseases and environmental impact. To decrease these problems and to reduce the amount of chemical fertilizer, the use of plant-growth-promoting rhizobacteria is an excellent alternative in potato production [1].

Plant-growth-promoting rhizobacteria are a group of bacteria isolated from the rhizosphere carrying several skills [2]. These skills are classified as direct when the action acts directly on the plant and indirect when the plant growth results from acting [3]. Some direct skills are phytohormone production, which promotes root development, improves soil exploitation efficiency, and increases water and nutrient absorption through plant roots [4]. Phytohormones also promote areal part increases, improving photosynthesis efficiency as a result of both improving plant growth and providing more metabolites to the rhizospheric microbial community through rhizodeposition. The phytohormones produced by some bacteria also protect the bacteria against the plant's protection mechanisms and decrease the bacteria's competition for their colonization niche [5]. Another direct skill is the capacity of some bacteria to provide important nutrients to plants. Some bacteria synthesize organic acids or enzymes that solubilize phosphorus in the soil [6,7]. Usually, there is a high amount of phosphorus in the soil; however, it is unavailable. These bacteria can solubilize this phosphorus, making it available to plants and other microorganisms. Some bacteria carry the enzyme named nitrogenase; this enzyme transforms atmospheric

nitrogen into ammonia, an assimilable form for many organisms [8]. These bacteria are called nitrogen fixers. There are some indirect skills that also promote plant growth, such as cellulolytic and amylolytic activity and siderophore production. These abilities make the carbon available and release energy into the soil. The capacity to produce siderophores makes iron available. Iron is present in the soil in small amounts, which is important for plants and microbes to grow.

This particular scenario of global agriculture and the possibility of using plant growth promoters drives many researchers worldwide to search for a deep understanding and to search for news isolates with these skills. Many studies have shown the benefits of using these bacteria, but new studies are welcome.

The *B. cereus*, *S. dextrinosolvens*, and *L. acidophilus* used in this study belong to the collection of the soil microbiology laboratory, and they were isolated from ruminants. The first study carried out with these bacteria was conducted by Santos et al. [9] in maize and soybean plants under greenhouse conditions. The present study used the same isolates characterized by Santos et al. [9] in potato plants under greenhouse conditions.

The objective of this study was to evaluate the use of three bacteria, *B. cereus*, *S. dextrinosolvens* and *L. acidophilus*, with some skills related to plant growth promotion on potato plants.

## 2. Materials and Methods

The experiment was carried out in a randomized block design in greenhouse conditions with five treatments and six repetitions with vases of 5 $dm^3$ with soil. The treatments were as follows: T1 = control; T2 = *B. cereus*; T3 = *L. acidophilus*; T4 = *S. dextrinosolvens*; T5 = "MIX" (mixture of *B. cereus* + *L. acidophilus* + *S. dextrinosolvens*).

### 2.1. Bacterial Isolates

The bacteria used in this study (*B. cereus*, *L. acidophilus*, and *S. dextrinosolvens*) belonged to the Laboratory of Soil Microbiology collection. These bacteria were grown in Erlenmeyer flasks containing nutrient broth medium at 28 °C for 24 h. The concentration was $1 \times 10^8$ $mL^{-1}$ colony-forming units (CFU).

### 2.2. Starch Agar

The following reagents/conditions were used to prepare the starch production medium: $K_2HPO_4$ 0.3 g $L^{-1}$; $MgSO_4.7H_2O$ 1.0 g $L^{-1}$; NaCl 0.5 g $L^{-1}$; $NaNO_3$ 1.0 g $L^{-1}$; starch 10 g $L^{-1}$; and pH = 6.9.

### 2.3. Cellulolytic Activity

The cellulolytic activity was assayed by monitoring the oxidation of L-3,4-dihydroxyphenylalanine (L-DOPA; Sigma, Sao Paulo Brazil) in hydrogen peroxide. A final volume of 1.0 mL of reaction mixture contained 4.0 mM hydrogen peroxide, 0.1 M potassium phosphate buffer (pH 7.0), and 1.0 mM L-DOPA. A concentrated crude enzyme preparation (100 to 200 μL) was used in the assay. The reaction was initiated by adding hydrogen peroxide, and the increase in the absorbance was monitored for 5 min at 37 °C. Responses containing all reagents except for the crude enzyme extract served as controls. One unit of the enzyme was expressed as the amount of enzyme. The culture medium methodology described by Ramachandra, Crawford, and Pometto with no alterations was used [10].

### 2.4. Production of Indoleacetic Acid

The bacteria evaluated were screened for IAA production. Briefly, the bacterial culture was inoculated in the respective medium (Jensen's/nutrient broth) with tryptophan (1, 2, and 5 mg $mL^{-1}$) or without tryptophan and incubated at 28 ± 2 °C for 15 days. Cultures were centrifuged at 3000 rpm for 30 min. Two milliliters of the supernatant was mixed with two drops of orthophosphoric acid and 4 mL of Solawaski's reagent (50 mL, 35% perchloric acid; 1 mL 0.5 $FeCl_3$). The development of a pink color indicates IAA production. The

reaction was read at 530 nm using a Spectronic 20D+. The level of IAA produced was estimated by a standard IAA graph.

### 2.5. P Quantification in Test Tubes

The quantification of phosphate solubilization was measured in a 120 mL Erlenmeyer flask containing 50 mL of Nahas medium [11]; 200 μL of inoculum from each isolate was added. Erlenmeyer flasks were incubated for 48 h at ± 28 °C with stirring at 180 rpm. After incubation, 5 mL of each sample was transferred to tubes and centrifuged at 9000 rpm for 15 min. Then, 1 mL of supernatant from each isolate, 4 mL of distilled water, and 1 mL of ammonium molybdate–vanadate reagent (formed by mixing equal volumes) were added to a new tube for further reading (after 5 min) in a spectrophotometer at 470 nm. To quantify the dynamics of the phosphorus levels, all the bacteria were grown in a minimal medium with 1 g of hydroxyapatite with three repetitions for each bacterium. Every three days, three tubes of each bacteria had their phosphorus content measured, and a graphic was created according to the results.

### 2.6. Nitrogen Quantification in Test Tubes

The method of Kuss [12] was used in the nitrogen quantification analyses by isolates. In determining nitrogen in foodstuffs, a digestion mixture of 40 g of sodium sulfate and 1.6 g of copper sulfate per 100 mL of acid is recommended, with a digestion time of 6 h. For the micro determination of protein in 50% glycerol, bromine is used as an oxidizing agent, supplemented by 30% hydrogen peroxide. To quantify the dynamics of phosphorus levels, all the bacteria were grown in a nitrogen-free medium with three repetitions for each bacterium. Every three days, three tubes of each bacteria had their nitrogen content measured, and a graphic was created according to the results.

### 2.7. Planting

Experiments were conducted in a greenhouse belonging to the Laboratory of Agricultural Microbiology of UNESP-FCAV (coordinates—Latitude: 21°14′05″ S Longitude: 48°17′09″ W). For studies of potatoes, the variety Dow Agro-Science was used. In experiments with potatoes, seeds of the variety Baroka Piornner were used. In both cases, seeds were inoculated with *B. cereus*, *L. acidophilus*, and *S. dextrinosolvens* deposited in pots (5 L) filled with red eutrophic latosol-type soil that was sieved and fertilized. Fertilization was performed according to soil chemical analysis and recommended for crops. The soil fertility composition was pH = 6.5; organic matter = 11 g dm$^3$; phosphorus = 20 mg dm$^3$; soil = 12 mg dm$^3$; potassium = 0.7 mmol$_c$ dm$^3$; magnesium = 17 mmol$_c$ dm$^3$; and the sum of bases = 24.4 mmol$_c$ dm$^3$.

### 2.8. Inoculations

Four inoculations were performed, the first through seeds immersed in 125 mL Erlenmeyer flasks containing nutrient broth at a bacterial concentration of $10^8$ CFU mL$^{-1}$ for 15 min in 120 rpm orbital shaking and then sown. The second, third, and fourth inoculations were performed every week, seven days after sowing, adding 20 mL of each inoculum at the same concentration as above.

## 3. Evaluations in Potato Plants

### 3.1. Dry Mass

Roots were collected from both cultures, washed in running water to remove excess soil, and dried on absorbent paper. Shoots were separated from roots, and both were dried in the oven with forced air circulation at 65 °C for approximately 72 h until reaching constant weight. The last step was the weighing of all the material using an analytical scale to determine the mass (g) of root dry matter (RDM) and shoot dry matter (SDM).

*3.2. Nitrogen Concentration in Shoots and Roots*

To determine the nitrogen concentration (N), the plant material was ground in a Willey mill (mesh 20) and submitted to N leaf analysis using the method proposed by Bremmer et al. [13].

*3.3. Shoot and Root Phosphorus Concentrations*

Phosphorus concentrations (P) were determined by nitroperchloric digestion, followed by the molybdovanadate colorimetric method according to the methodology proposed by Haag et al. [14].

*3.4. Statistical Analysis*

The means were compared using Duncan's test with a 5% probability using the software Agroestat [15].

**4. Results**

The bacterium *B. cereus* showed the ability to synthesize amylase, IAA production of 9.08 $\mu$g mL$^{-1}$, phosphorus solubilization of 14.93 mg P L$^{-1}$, and nitrogen fixation of 0.7 mg of N L$^{-1}$. *S. dextrinosolvens* showed the ability to synthesize siderophores and amylase, IAA production of 10.25 $\mu$g mL$^{-1}$, phosphorus solubilization of 41.38 mg P L$^{-1}$, and nitrogen fixation of 0.42 mg N L$^{-1}$. *L. acidophilus* showed the ability to synthesize siderophores, IAA production of 7.25 $\mu$g mL$^{-1}$, phosphorus solubilization of 5.58 mg P L$^{-1}$, and nitrogen fixation of 0.5 mg N L$^{-1}$ (Table 1).

**Table 1.** The presence of siderophores, amylolytic and cellulolytic activities, IAA, phosphorus solubilization, and nitrogen fixation were shown by the bacteria *B. cereus*, *S. dextrinosolvens*, and *L. acidophilus*. The values were measured by Santos et al. (2020) [9].

| Isolate | Siderophores | Amylolytic Activity | Cellulolytic Activity | IAA $\mu$g mL$^{-1}$ | P Solubilization mg P L$^{-1}$ | N Fixation mg N L$^{-1}$ |
|---------|:---:|:---:|:---:|:---:|:---:|:---:|
| *B. cereus* | - | + | - | 9.08 | 14.93 | 0.70 |
| *S. dextrinosolvens* | + | + | - | 10.25 | 41.38 | 0.42 |
| *L. acidophilus* | + | - | - | 7.25 | 5.58 | 0.50 |

*Dry Matter*

The highest value for shoot dry matter (SDM) was found for *B. cereus* (T2), followed by Mix (T5) ($p < 0.05$). There was no significant difference between the other treatments ($p > 0.05$) (Figure 1). For the root dry matter (RDM), the highest value was found for *L. acidophilus* (T3), followed by Mix (T5) ($p < 0.05$). There was no significant difference between the other treatments ($p > 0.05$) (Figure 1). For the nitrogen content from the shoot, there was no significant difference between the treatments ($p > 0.05$). For the nitrogen content from the root, there was no significant difference between the treatments ($p > 0.05$) (Figure 2).

There were no significant differences between the treatments for phosphorus content in the shoot ($p > 0.05$). The highest phosphorus contents from the root were found in *L. acidophilus* (T3), *S. dextrinosolvens* (T4), and Mix (T5) ($p < 0.05$). There was no significant difference between *B. cereus* (T2) and the control (T1) ($p > 0.05$) (Figure 3).

Figure 4 depicts the dynamics of the phosphorus content over 12 days. Interestingly, the behaviors of all bacteria were similar, with a few differences. For the first three days, the solubilized phosphorus content increased until day nine for *S. dextrinosolvens*, and then decreased until day 12. This behavior was similar to that of *L. acidophilus*. For *B. cereus*, from three days, the phosphorus content decreased quickly and started increasing up to 12 days. This dynamic is controlled by releasing enzymes or organic acids and solubilizing phosphorus and making it available. The reduction in phosphorus content occurs by

absorbing phosphorus for microorganisms to grow. The mix of bacteria increased the phosphorus content up to day six and decreased it to day 12.

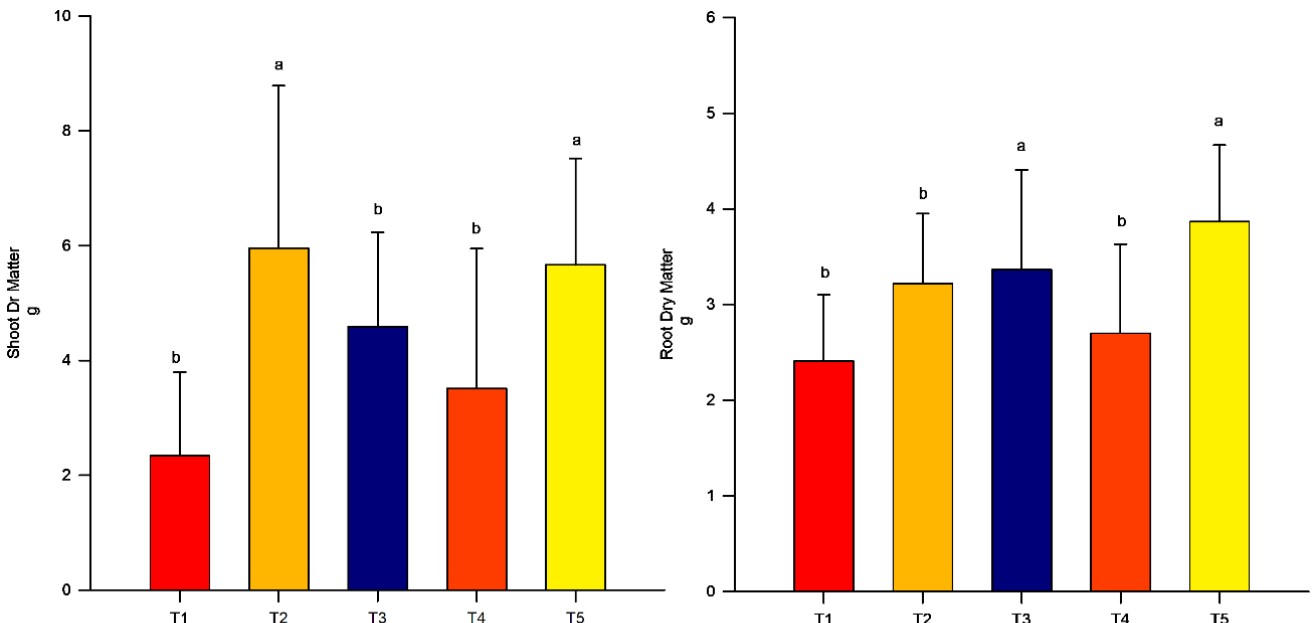

**Figure 1.** Shoot and root dry matter of potato plants inoculated with different bacteria. Bars represent the standard error of the mean. T1 = control; T2 = *B. cereus*; T3 = *L. acidophilus*; T4 = *S. dextrinosolvens*; T5 = Mix. Means with different letters indicate significant differences among treatments. Statistical analysis was performed using Duncan's test ($p \leq 0.05$). Letter a indicates the highest value. Letter b indicates the second highest value.

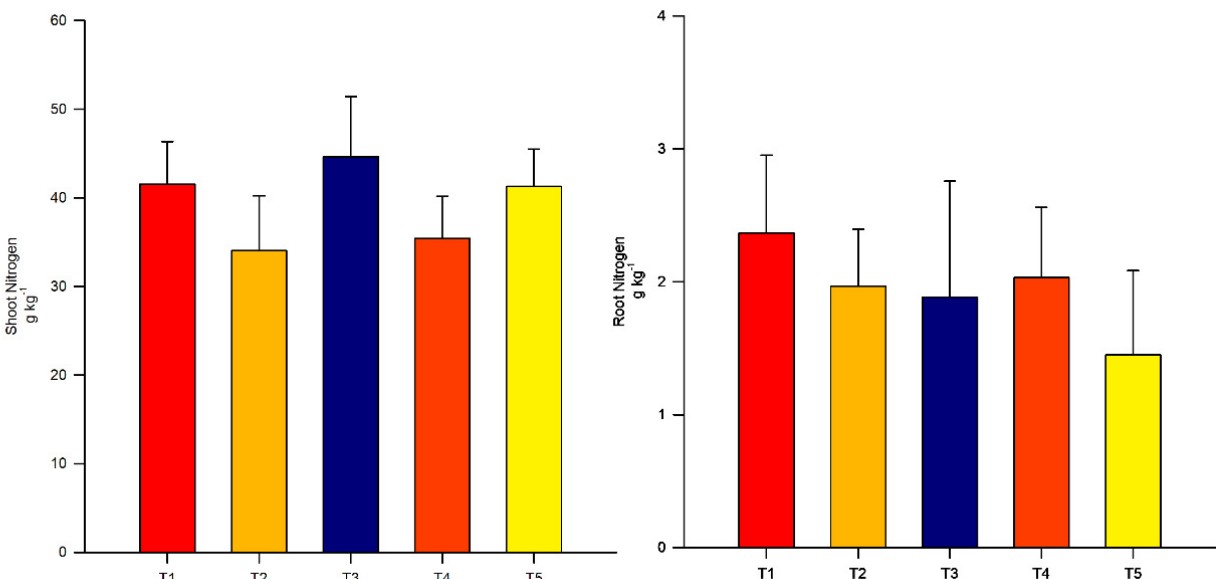

**Figure 2.** Nitrogen content from the shoots and roots of potato plants inoculated with different bacteria. Bars represent the standard error of the mean. T1 = control; T2 = *B. cereus*; T3 = *L. acidophilus*; T4 = *S. dextrinosolvens*; T5 = Mix.

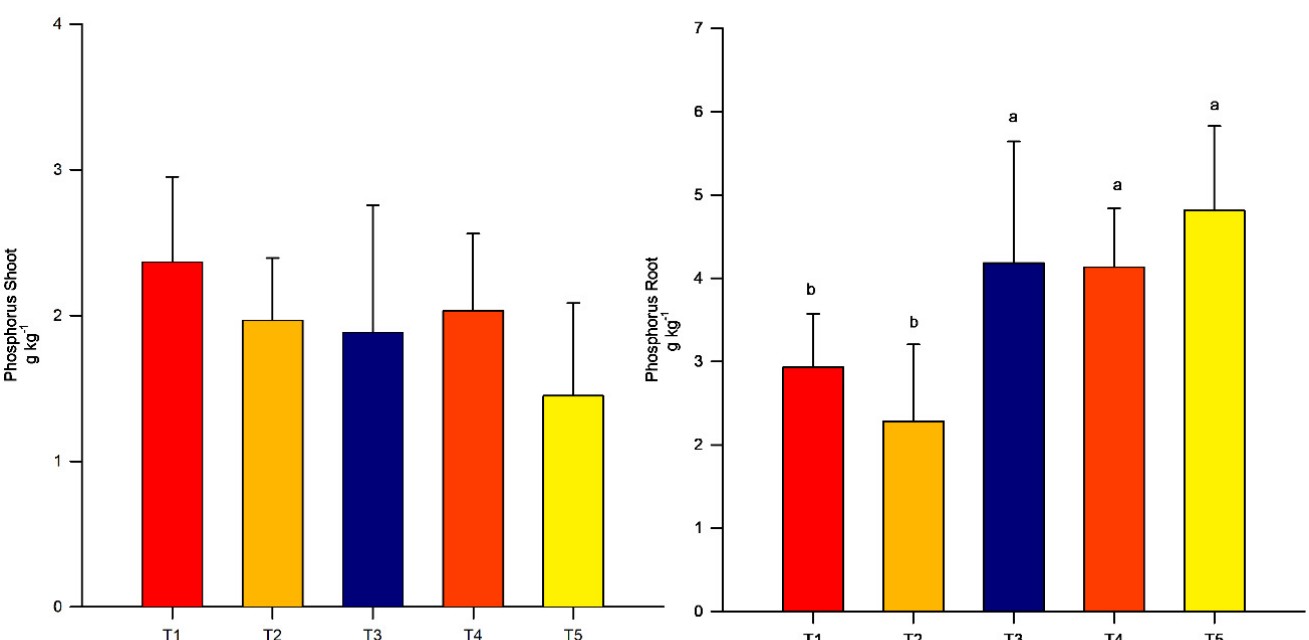

**Figure 3.** Phosphorus content from the shoots and roots of potato plants inoculated with several bacteria. T1 = control; T2 = *B. cereus*; T3 = *L. acidophilus*; T4 = *S. dextrinosolvens*; T5 = Mix. Means with different letters indicate significant differences among treatments. Statistical analysis was performed using Duncan's test ($p \leq 0.05$). Letter a indicates the highest value. Letter b indicates the second highest value.

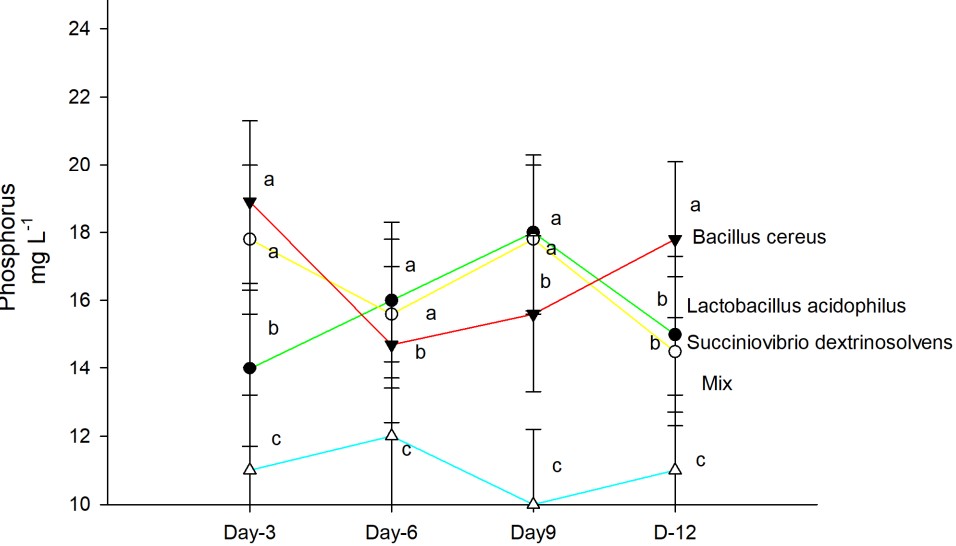

**Figure 4.** Dynamic of solubilized phosphorus content from the test tube on three, six, nine, and twelve days with three repetitions for each bacterium every three days. Means with different letters indicate significant differences among treatments. Statistical analysis was performed using Duncan's test ($p \leq 0.05$). Black triangle = *B. cereus*; Black cicle= *L. acidophilus*; white circle= *S. dextrinosolvens*; white triangle = Mix.

Figure 5 depicts the dynamics of nitrogen content in the test tube. Interestingly, the bacterium *B. cereus* showed the highest nitrogen content for the first three days, and it decreased up to day nine and increased to 12 days. The behaviors of *L. acidophilus*, *S. dextrinosolvens*, and the mixture were similar. Interestingly, the mixture containing the bacterium *B. cereus* did not behave the same way as when *B. cereus* was alone. For 12 days, the nitrogen content from *B. cereus* and *L. acidophilus* was higher than that from *S. dextrinosolvens* and Mix.

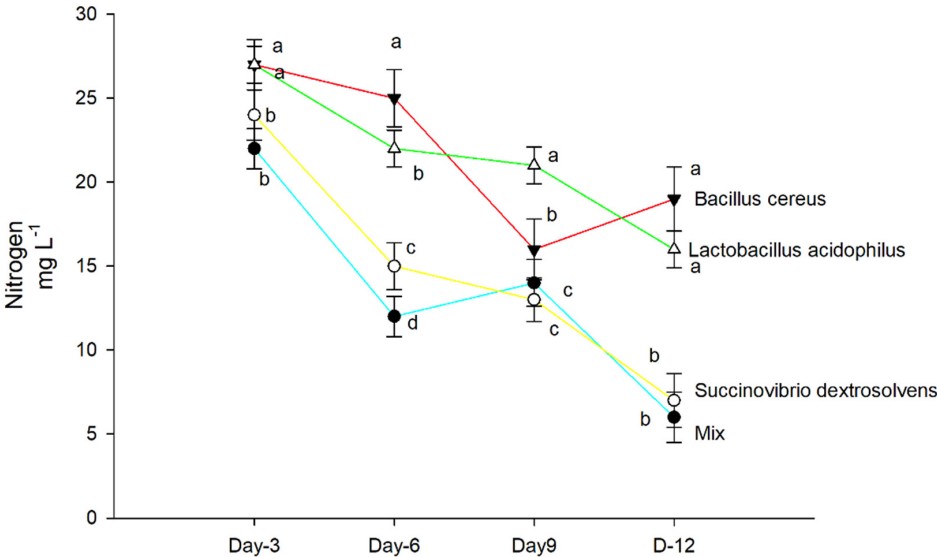

**Figure 5.** Dynamic of solubilized phosphorus content from the test tube on three, six, nine, and twelve days with three repetitions for each bacterium every three days. Means with different letters indicate significant differences among treatments. Statistical analysis was performed using Duncan's test ($p < 0.05$). Black triangle = *B. cereus*; Black cicle= *L. acidophilus*; white circle = *S. dextrinosolvens*; white triangle = Mix.

The highest phosphorus content was found for day three when the bacterium *B. cereus* was inoculated with *S. dextrinosolvens* (T1 + T2). There was a quick reduction, and then the phosphorus content increased until day 12. The inoculation of *S. dextrinosolvens* with *L. acidophilus* (T2 + T3) promoted an increase in phosphorus content up to day nine and decreased it up to day 12. The inoculation of *B. cereus* with *L. acidophilus* (T1 + T3) increased the phosphorus content from day three up to day nine, and then the phosphorus content was reduced up to day 12. For nitrogen content, inoculation of *B. cereus* with *S. dextrinosolvens* promoted an increase on day three, and the values remained the same up to day 12. For inoculation with *S. dextrinosolvens* and *L. acidophilus*, the nitrogen content increased on day six and remained the same until day 12. The nitrogen content was the same throughout the trial for the inoculation of *B. cereus* with *L. acidophilus* (Figure 6).

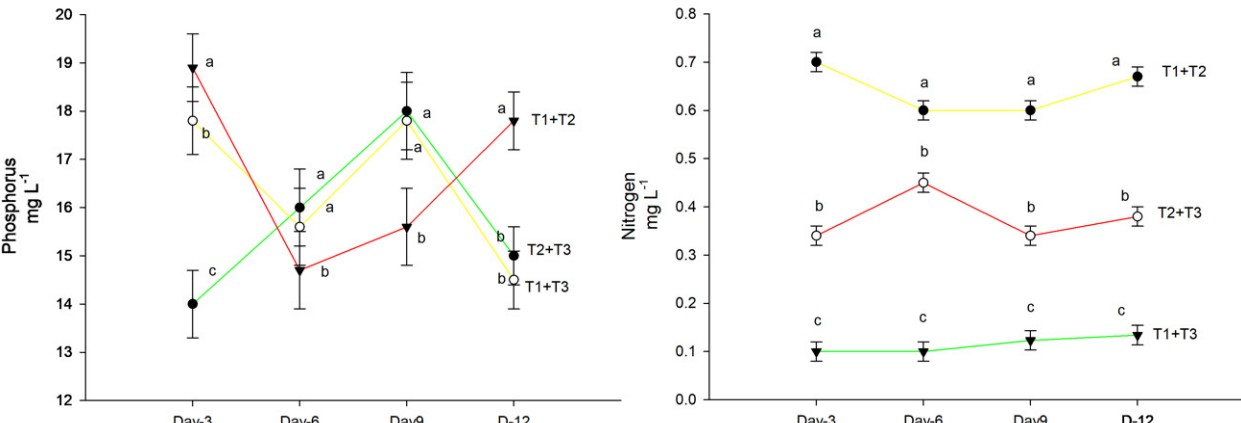

**Figure 6.** Dynamic of solubilized phosphorus and fixed nitrogen content from the test tube every three days with the mixture of bacteria. T1 = *B. cereus*; T2 = *S. dextrinosolvens*, and T3 = *L. acidophilus*. Means with different letters indicate significant differences among treatments. Statistical analysis was performed using Duncan's test ($p < 0.05$). Black triangle = T1 + T2; black circle = T2 + T3; white circle = T1 + T3.

## 5. Discussion

The bacteria *B. cereus*, *S. dextrinosolvens*, and *L. acidophilus* were demonstrated to be carriers of some abilities related to plant growth promotion, such as the presence of siderophores, cellulosic activity, amylolytic activity, phytohormone production (particularly the production of auxin), phosphorus solubilization, and nitrogen fixation. Several studies have shown the utilization of *B. cereus* and *S. dextrinosolvens* in animal feed. Many studies have demonstrated the benefits of using the bacterium *L. acidophilus* in animal and human feed. However, Dos Santos carried out the first report about the characterization of abilities and their utilization for promoting plant growth [9]. This pioneering study opened an avenue to the realization of other studies in the incessant search to find bacteria that promote plant growth and development, collaborating with sustainable production because of lower production costs and environmental impact without a reduction in productivity. The fact that these bacteria are carriers of abilities related to plant growth justifies the realization of studies to verify whether these bacteria promote plant growth. These microorganisms improve nutrient uptake, nutrient use efficiency, abiotic stress tolerance, biocontrol, and the quality of agricultural production.

Aljeboury and Mahmoud [16] verified the antagonistic effect of four isolates of bacterial producers of lactic acid, including the bacterium *L. acidophilus*, against pathogenic bacteria such as *Escherichia coli*, *Klebsiella*, and *Salmonella*. Interestingly, this study also verified the antagonistic effect with a reduction in phytopathogenic fungal growth that causes much damage to agricultural crops. These fungi were *Alternaria* sp. and *Fusarium oxysporium*. These fungi cause diseases in several agricultural crops of large economic importance, such as tomato, cauliflower, potato, citrus, and several ornamental plants. *Alternaria* fungi are also responsible for synthesizing secondary metabolites such as aflatoxin. This study found that the antagonistic effect provoked by *L. acidophilus* occurs through the synthesis of several substances, such as organic acids, hydrogen peroxide, and bacteriocins [16,17]. Sharf et al. [18] assessed the potential of two species of plant-growth-promoting rhizobacteria (PGPR), viz. *Bacillus megaterium* and *Pseudomonas fluorescence*, and an allelopathic weed, *Anagallis arvensis* L., for the control of southern blight disease in chilies. Some studies carried out by [19–22] show that some lactic bacteria may regulate the soil organic matter and the biochemical cycles and improve plant health by reducing soil contamination. Zang et al. [23] verified that using *L. plantarium*, *L. acidophilus*, and *L. rhamnosus* inoculated alone or in combination reduced soil contamination by cadmium and zinc and promoted the plant growth of mustard (*Brassica juncea*), increasing biomass and chlorophyll content. El-Saadory [24] verified that the use of *L. acidophilus* controlled the diseases caused by *Fusarium graminearum* in wheat and promoted an increase in biomass and productivity.

There are some inconsistencies in the results with the use of plant-growth-promoting microorganisms. This may be due to the microorganism's inefficiency in colonizing the plant and its rhizosphere. It happens when the plant does not interact with the microorganisms. To increase the chance of the plant interacting with the microorganisms, a good strategy is to use more than one isolate; if there is a synergistic effect between the microorganisms, their abilities may be increased. Therefore, the combination of microorganisms was evaluated (T1 + T2), (T2 + T3), and (T1 + T3). The results showed no increase in their abilities to solubilize phosphorus and fix nitrogen. However, the mixture of microorganisms showed some positive effects.

## 6. Conclusions

Lactic bacteria, as plant-growth-promoting bacteria, show great potential for their use since *L. acidophilus* increases the dry matter in potato plants. In the future, these lactic bacteria could be used in agricultural production as inoculates, significantly reducing mineral fertilizer levels and contributing to more sustainable production.

**Author Contributions:** Conceptualization, E.C.R. and L.D.P.; methodology, L.D.P. and J.D.; software, C.H.B.S.; validation, E.T.F., L.R.S. and L.A.d.A.; formal analysis, L.D.P.; investigation, L.D.P.; resources, E.C.R.; data curation, L.D.P.; writing—original draft preparation, L.D.P.; writing—review and editing, L.D.P. and E.C.R.; visualization, E.C.R.; supervision, E.C.R.; project administration, E.C.R.; funding acquisition, E.C.R. All authors have read and agreed to the published version of the manuscript.

**Funding:** This research received no external funding.

**Institutional Review Board Statement:** Not applicable.

**Informed Consent Statement:** Not applicable.

**Data Availability Statement:** Not applicable.

**Acknowledgments:** The authors thank CAPES by scholarship code: 001.

**Conflicts of Interest:** The authors declare no conflict of interest.

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
