# Peer review of "Lactic Bacteria with Plant-Growth-Promoting Properties in Potato"

_2036-7481, doi:10.3390/microbiolres14010022_

Round 1
Reviewer 1 Report
In general, the research work is good. However, substantial revision is required for the improvement of this manuscript before its publication.
1- Title should be changed to "Lactic Acid Producing Bacteria with Plant Growth Promoting Potential in Potato".
2- Rewrite the Abstract in a better scientific language. Reduce the length of initial introductory portion. Write a more impressive concluding sentence at the end of abstract. Also improve language of the whole abstract.
3- Arrange the keywords alphabetically and also add a few more.
4- Improve language and formatting throughout the manuscript. Especially write units uniformly.
5- Avoid too many short paragraphs in Introduction. Combine related paragraphs to form maximum of 2 or 3 paragraphs for better presentation and understanding. Combine paragraphs 2 and 3. Similarly, combine paragraphs 3 and 4. Delete paragraph 5 as it is repetition of paragraph 4.
6- Some sentences in the Introduction are without any reference. I have suggested a few references but many more are required.
My suggested references are given here. I have mentioned their positions in the Introduction. Authors should check their suitability.
Sharf W, Javaid A, Shoaib A, Khan IH (2021). Induction of resistance in chili against Sclerotium rolfsii by plant growth promoting rhizobacteria and Anagallis arvensis. Egyptian Journal of Biological Pest Control 31: 16.
Javed S, Javaid A, Hanif U, Bahadur S, Sultana S, Shuaib M, Ali S (2021). Effect of necrotrophic fungus and PGPR on the comparative histochemistry of Vigna radiata by using multiple microscopic techniques. Microscopy Research and Technique 84(11): 2737-2748.
Javaid A, Bajwa R (2011). Effect of effective microorganism application on crop growth, yield and nutrition in Vigna radiata (L.) Wilczek in different soil amendment systems. Communications in Soil Science and Plant Analysis 42(17): 2112-2121.
7- Write formulas and units correctly in Methodology. Also write Methodology in more scientific language.
8- Improve Discussion. Avoid short paragraphs.
9- Stats in the Figures seems incorrect. There are very large standard errors values which are overlapped but there is significant differences among the treatments.
10- Format References uniformly and correctly.

Author Response
Reviewer1 - In general, the research work is good. However, substantial revision is required for the improvement of this manuscript before its publication.
Answer: We thank the reviewer for the opportunity to given to us to improve our manuscript.
Reviewer1 - 1- Title should be changed to "Lactic Acid Producing Bacteria with Plant Growth Promoting Potential in Potato".
Answer: It has been done
Reviewer1 - 2- Rewrite the Abstract in a better scientific language. Reduce the length of initial introductory portion. Write a more impressive concluding sentence at the end of abstract. Also improve language of the whole abstract.
Answer: It has been done.
Reviewer1 - 3- Arrange the keywords alphabetically and also add a few more.
Answer: It has been done
Reviewer1 - 4- Improve language and formatting throughout the manuscript. Especially write units uniformly.
Answer: It has been done
Reviewer1 - 5- Avoid too many short paragraphs in Introduction. Combine related paragraphs to form maximum of 2 or 3 paragraphs for better presentation and understanding. Combine paragraphs 2 and 3. Similarly, combine paragraphs 3 and 4. Delete paragraph 5 as it is repetition of paragraph 4.
Answer: It has been done
Reviewer1 - 6- Some sentences in the Introduction are without any reference. I have suggested a few references but many more are required.
Answer: It has been done.
Reviewer1 - My suggested references are given here. I have mentioned their positions in the Introduction. Authors should check their suitability.
Answer: Some references have been added.
Sharf W, Javaid A, Shoaib A, Khan IH (2021). Induction of resistance in chili against Sclerotium rolfsii by plant growth promoting rhizobacteria and Anagallis arvensis. Egyptian Journal of Biological Pest Control 31: 16.
Javed S, Javaid A, Hanif U, Bahadur S, Sultana S, Shuaib M, Ali S (2021). Effect of necrotrophic fungus and PGPR on the comparative histochemistry of Vigna radiata by using multiple microscopic techniques. Microscopy Research and Technique 84(11): 2737-2748.
Javaid A, Bajwa R (2011). Effect of effective microorganism application on crop growth, yield and nutrition in Vigna radiata (L.) Wilczek in different soil amendment systems. Communications in Soil Science and Plant Analysis 42(17): 2112-2121.
Reviewer1 - 7- Write formulas and units correctly in Methodology. Also write Methodology in more scientific language.
Reviewer1 - 8- Improve Discussion. Avoid short paragraphs.
Reviewer1 - 9- Stats in the Figures seems incorrect. There are very large standard errors values which are overlapped but there is significant differences among the treatments.
Reviewer1 - 10- Format References uniformly and correctly.

Reviewer 2 Report
In my opinion the article needs majore improvements.
Some examples of improvements are presented below:
- The abstract should not have more than 200 words.
- Please explain the acronym when it appears for the first time in the text (e.g. IAA – abstract)
- I suggest to replace the „nitrogen per liter” with NžL-1 (abstract, row 10).
- I suggest to replace P L-1 , N L-1, μg mL-1 with PžL-1 , NžL-1, μgžmL-1 . Please look for this aspect in the whole article.
- Keywords: the microorganisms name in italic please. Please look for this aspect in the whole article.
- Spaces between words: I suggest checking the entire article, looking at the spaces between words.
- Please write the names of the microorganisms either using the full name (e.g.Bacillus cereus) or the short one (e.g. B. cereus). / Please look for this aspect in the whole article for all microorganisms.
- I suggest you pay attention to the punctuation and the spaces between words.
- Please add the correct formulas, e.g. K2HPO4 - K2HPO4 . Please look for this aspect in the whole article for all chemical formulas.
- I suggest to replace g/L with gžL-1 . Please look for this aspect in the whole article.
- in hydrogen peroxide (28) - Is (28) necessary?
- I suggest to replace μl with μL, ml with mL . Please look for this aspect in the whole article.
- A470? Please explain.
- production (15) - Is (15) necessary?
- O.D.? Please explain.
- Phosphorus quantification in test tubes
- I suggest to replace (1997) with [1997].
- (1994)? Is (1994) necessary?
- The method of [15]. Pleas add the author name before [15].
- foodstuffs (1)? - Is (1) necessary?
- (51)? - Is (51) necessary?
- 7 mmolc dm3; magnesium 17 mmolc dm3; and the sum of bases 24.4 mmolc dm3 ?
- 108 CFU ml -1?
- proposed by [16]. Pleas add the author name before [16].
- proposed by [17]. Pleas add the author name before [17].
- I suggest to improve the Table 1.
- The name of the table must be placed above, not below.
- References should be prepared in accordance with the requirements of the journal (https://www.mdpi.com/journal/microbiolres/instructions).
Author Response
Reviewer2 In my opinion the article needs major improvements.
Answer: We thank the reviewer for the opportunity given to us to improve our manuscript
Some examples of improvements are presented below:
Reviewer2- The abstract should not have more than 200 words.
Answer: It has been reduced
Reviewer2- Please explain the acronym when it appears for the first time in the text (e.g. IAA – abstract)
Answer: IAA means indole acetic acid – It has been added.
Reviewer2- I suggest to replace the „nitrogen per liter” with NžL-1 (abstract, row 10).
Answer: It has been replaced.
Reviewer2- I suggest to replace P L-1 , N L-1, μg mL-1 with PžL-1 , NžL-1, μgžmL-1 . Please look for this aspect in the whole article.
Answer: Sorry, I did not understand the difference. Did you suggest double space?
Reviewer2- Keywords: the microorganisms name in italic please. Please look for this aspect in the whole article.
Answer: The names of microorganisms have been written in italic. I do not know why in the Journal’s system they have been changed. Please verify the pdf file.
Reviewer2 Spaces between words: I suggest checking the entire article, looking at the spaces between words.
Answer: It has been checked
Reviewer2 Please write the names of the microorganisms either using the full name (e.g.Bacillus cereus) or the short one (e.g. B. cereus). / Please look for this aspect in the whole article for all microorganisms.
Answer: It has been checked,
Reviewer2- I suggest you pay attention to the punctuation and the spaces between words.
Answer: It has been checked
Reviewer2 Please add the correct formulas, e.g. K2HPO4 - K2HPO4 . Please look for this aspect in the whole article for all chemical formulas.
Answer: It has been changed
Reviewer2 I suggest to replace g/L with gžL-1 . Please look for this aspect in the whole article.
Answer: it has been changed
Reviewer2 in hydrogen peroxide (28) - Is (28) necessary?
Answer: It has been removed
Reviewer2 I suggest to replace μl with μL, ml with mL . Please look for this aspect in the whole article.
Answer: it has been changed.
Reviewer2 A470? Please explain.
Answer: It has been changed for absorbance.
Reviewer2 production (15) - Is (15) necessary?
Answer: It has been removed
- Reviewer2 O.D.? Please explain.
Answer: It has been removed
Reviewer2 Phosphorus quantification in test tubes
Answer: It has been changed
Reviewer2 I suggest to replace (1997) with [1997].
Answer: It has been removed.
Reviewer2 (1994)? Is (1994) necessary?
Answer: It has been removed
Reviewer2 The method of [15]. Pleas add the author name before [15].
Answer: It has been added
Reviewer2 foodstuffs (1)? - Is (1) necessary?
Answer: It has been removed.
- Reviewer2 (51)? - Is (51) necessary?
Answer: It has been removed.
Reviewer2 7 mmolc dm3; magnesium 17 mmolc dm3; and the sum of bases 24.4 mmolc dm3 ?
Answer: These values are the soil fertility.
Reviewer2 108 CFU ml -1?
Answer: It has been changed for 108
Reviewer2 proposed by [16]. Pleas add the author name before [16].
Answer: It has been added
Reviewer2 proposed by [17]. Pleas add the author name before [17].
Answer:
Reviewer2 I suggest to improve the Table 1.
Answer: It has been done.
- Reviewer2 The name of the table must be placed above, not below.
Answer: It has been made.
- References should be prepared in accordance with the requirements of the journal (https://www.mdpi.com/journal/microbiolres/instructions).

Reviewer 3 Report
The paper is interesting and within the scope of the journal.
-The authors should improve the introduction by adding information about the mechanisms involved with bacteria and Plant Growth.
_Figures should be revised. Legends should be added, and the XX axis should be uniformized.
-Conclusion should be improved
-The references in the reference section should be revised following the journal rules.
Author Response
Reviewer3 - The paper is interesting and within the scope of the journal.
Answer: Thank the reviewer for the opportunity given to us to improve our manuscript.
Reviewer3 -The authors should improve the introduction by adding information about the mechanisms involved with bacteria and Plant Growth.
Reviewer3_Figures should be revised. Legends should be added, and the XX axis should be uniformized.
Answer: It has been done. Please see the pdf file.
Reviewer3Conclusion should be improved
Answer:
Reviewer3-The references in the reference section should be revised following the journal rules.
Answer: It has been done.

Reviewer 4 Report
In this study, author aimed to evaluate the abilities of three bacteria, Bacillus cereus, Succinovibrio dextrinosolvens, and Lactobacillus acidophilus, to fix nitrogen, solubilize phosphorus and produce cellulosic and amylolytic enzymes in potato. B. cereus showed great potential to be used as a plant growth promoter in potato crops in the future. The article is not well organized. Finally, there are some essential problems should be addressed by authors, which are listed below.
1. There is no line number, which is unfriendly for reviewer to give comments.
2. The title of this paper is Lactic Bacteria with Plant Growth-Promoting Potential, but the abstract and introduction are about potatoes, which does not fit the title.
3.P1 “Potato crop feed many people worldwide, require a lot of chemical fertilizers that are a renewable source, and may cause environmental impact.” The sentence doesn’t flow and clear. Rewrite it.
4. P1 The logic of the first Introduction paragraph is not smooth, please modify it.
5. In the abstract and Instruction, sometimes the growth of plants is said, sometimes the growth of potatoes is said, the meaning is not clear enough, this paper only obtains the results in potato, cannot rise to all plants, please revise the manuscript.
6. P2 “Bacterial Isolates” part: “The” is not bold, delete the green background.
7. P2 “Starch agar” part: K2HPO4 instead of K2HPO4. Please revise the full text.
8. Three treatments can be added, B. cereus + L. acidophilus, L. acidophilus + S. dextrinosol-vens, B. cereus + S. dextrinosol-vens.
9. “Planting” part: dm3 instead of dm3, please check the superscript and subscript of the manuscript.
10. “Dry matter” part: p < 0.05 instead p<0.05, please revise the manuscript.
11. Specific conditions of potato growth under different treatments can be added, such as plant height, germination speed, potato yield, etc.
12. CONCLUSION part: base on the results of the article, the conclusions drawn are overstate.
Author Response
Reviewer4 There is no line number, which is unfriendly for reviewer to give comments.
Answer: It has been added.
Reviewer4 - 2. The title of this paper is Lactic Bacteria with Plant Growth-Promoting Potential, but the abstract and introduction are about potatoes, which does not fit the title.
Answer: The title has been changed to “Lactic Bacteria with Plant Growth-Promoting on Potato.
Reviewer4 - 3.P1 “Potato crop feed many people worldwide, require a lot of chemical fertilizers that are a renewable source, and may cause environmental impact.” The sentence doesn’t flow and clear. Rewrite it.
Answer: It has been removed.
Reviewer4 - 4. P1 The logic of the first Introduction paragraph is not smooth, please modify it.
Answer: It has been done.
Reviewer4 - 5. In the abstract and Instruction, sometimes the growth of plants is said, sometimes the growth of potatoes is said, the meaning is not clear enough, this paper only obtains the results in potato, cannot rise to all plants, please revise the manuscript.
Reviewer4 - 6. P2 “Bacterial Isolates” part: “The” is not bold, delete the green background.
Answer: It was a Journal´s system error. Please see the pdf file. It does not exist.
Reviewer4 - 7. P2 “Starch agar” part: K2HPO4 instead of K2HPO4. Please revise the full text.
Answer: It has been changed.
Reviewer4 - 8. Three treatments can be added, B. cereus + L. acidophilus, L. acidophilus + S. dextrinosol-vens, B. cereus + S. dextrinosol-vens.
Answer: Sorry, I did not understand.
Reviewer4 - 9. “Planting” part: dm3 instead of dm3, please check the superscript and subscript of the manuscript.
Answer: It has been checked and corrected.
Reviewer4 - 10. “Dry matter” part: p < 0.05 instead p<0.05, please revise the manuscript.
Answer: It has been changed.
Reviewer4 - 11. Specific conditions of potato growth under different treatments can be added, such as plant height, germination speed, potato yield, etc.
Answer:
Reviewer4 - 12. CONCLUSION part: base on the results of the article, the conclusions drawn are overstate.
Answer: It has been changed.

Round 2
Reviewer 1 Report
Authors failed to incorporate the suggested and highly necessary corrections.
1- Title should be "Lactic Acid Bacteria with Plant Growth-Promoting Potential in Potato". Authors should also change the title of submission portal of the journal.
2- Introduction is still with many small paragraphs of incomplete information.
3- Discussion is still very poor with many small paragraphs. Do not write in the form of such small paragraphs. Combine references in larger paragraphs having the same theme.
4- References needs correct formatting.
5- Figures are either without stats or with incorrect stats. Authors should recheck the stats. Bars (values) with such long and overlapped standard errors can be significantly different from one another.

Author Response
We thank the reviewer for his patience and the opportunity given to us to improve our manuscript.
Reviewer1- Title should be "Lactic Acid Bacteria with Plant Growth-Promoting Potential in Potato". Authors should also change the title of submission portal of the journal.
Answer: It has been changed.
Reviewer1 2- Introduction is still with many small paragraphs of incomplete information.
Answer: Many paragraphs were joined.
Reviewer1 - 3- Discussion is still very poor with many small paragraphs. Do not write in the form of such small paragraphs. Combine references in larger paragraphs having the same theme.
Answer: Many paragraphs were joined.
4- References needs correct formatting.
Answer: For reference, I used Endnote with the appropriate style for the Journal. I do not know what I need to change.
5- Figures are either without stats or with incorrect stats. Authors should recheck the stats. Bars (values) with such long and overlapped standard errors can be significantly different from one another.
Answer: The statistical analysis was added to figures 4-6.

Reviewer 2 Report
In my opinion, the article should be published.
However, relate to:
- I suggest to replace P L-1 , N L-1, μg mL-1 with PžL-1 , NžL-1, μgžmL-1 . Please look for this aspect in the whole article. Between the two letters should appear dot.
- 7 mmolc dm3; magnesium 17 mmolc dm3; and the sum of bases 24.4 mmolc dm3? dm3 or
dm-3?
- Row 159 - Bremmer et al., [13]. Please remove the comma.
- Row 163 - Haag et al., [14].Please remove the comma.
- In Table 1, the decimal numbers should have the same number of digits after comma. Example, in the last column, row 2 - 0.7 becomes 0.70; row 4 - 0.5 becomes 0.50.
- Row - 291 Sharf et al., [18]. Please remove the comma.
- I suggest you pay attention to the punctuation and the spaces between words. Exempli gratia - Row – 293 – Please add point at the end of the sentence.
Author Response
However, relate to:
I suggest to replace P L-1 , N L-1, μg mL-1 with PžL-1 , NžL-1, μgžmL-1 . Please look for this aspect in the whole article. Between the two letters should appear dot.
Answer: It has been done. Please see the pdf file.
- 7 mmolc dm3; magnesium 17 mmolc dm3; and the sum of bases 24.4 mmolc dm3? dm3 or
dm-3?
Answer: The correct if dm3. It has been changed. Maybe the Journal’s system has changed it for dm3. Please see the pdf file.
- Row 159 - Bremmer et al., [13]. Please remove the comma.
Answer: The comma has been removed.
- Row 163 - Haag et al., [14].Please remove the comma.
Answer: The comma has been removed.
- In Table 1, the decimal numbers should have the same number of digits after comma. Example, in the last column, row 2 - 0.7 becomes 0.70; row 4 - 0.5 becomes 0.50.
Answer: It has been added.
- Row - 291 Sharf et al., [18]. Please remove the comma.
- I suggest you pay attention to the punctuation and the spaces between words. Exempli gratia - Row – 293 – Please add point at the end of the sentence.
Answer: It has been added.

Reviewer 4 Report
Nice work
Author Response
We thank the reviewer for allowing us to improve our manuscript.
